# Black Phosphorus–Tungsten Oxide Sandwich-like Nanostructures for Highly Selective NO_2_ Detection

**DOI:** 10.3390/s24051376

**Published:** 2024-02-21

**Authors:** Canda Zheng, Yunbo Shi, Bolun Tang, Jianhua Zhang

**Affiliations:** Higher Educational Key Laboratory for Measuring & Control Technology and Instrumentations of Heilongjiang Province, Harbin University of Science and Technology, Harbin 150080, China; zhengcanda@outlook.com (C.Z.); tangbolun@sina.com (B.T.); zhangjianhua0919@163.com (J.Z.)

**Keywords:** black phosphorus, tungsten oxide, one-step hydrothermal method, sandwich-like nanostructures, micro-electro-mechanical system, selectivity, nitrogen dioxide

## Abstract

Modern chemical production processes often emit complex mixtures of gases, including hazardous pollutants such as NO_2_. Although widely used, gas sensors based on metal oxide semiconductors such as WO_3_ respond to a wide range of interfering gases other than NO_2_. Consequently, developing WO_3_ gas sensors with high NO_2_ selectivity is challenging. In this study, a simple one-step hydrothermal method was used to prepare WO_3_ nanorods modified with black phosphorus (BP) flakes as sensitive materials for NO_2_ sensing, and BP-WO_3_-based micro-electromechanical system gas sensors were fabricated. The characterization of the as-prepared BP-WO_3_ composite through X-ray diffraction scanning electron microscopy and X-ray photoelectron spectroscopy confirmed the successful formation of the sandwich-like nanostructures. The result of gas-sensing tests with 2–14 ppm NO_2_ indicated that the sensor response was 1.25–2.21 with response–recovery times of 36 and 36 s, respectively, at 190 °C. In contrast to pure WO_3_, which exhibited a response of 1.07–2.2 to 0.3–5 ppm H_2_S at 160 °C, BP-WO_3_ showed almost no response to H_2_S. Thus, compared with pure WO_3_, BP-WO_3_ exhibited significantly improved NO_2_ selectivity. Overall, the BP-WO_3_ composite with sandwich-like nanostructures is a promising material for developing highly selective NO_2_ sensors for practical applications.

## 1. Introduction

With the rapid development of modern industry, various health-threatening atmospheric pollutants have emerged. A representative example is the inorganic compound NO_2_, a toxic gas with an irritating odor [1]. NO_2_ inhalation can lead to interstitial lung lesions, respiratory infections, and lung fibrosis [2,3]. Anthropogenic NO_2_ mainly originates from high-temperature combustion processes and is released through fuel combustion, motor vehicle exhaust, and boiler exhaust emissions [4]. In practice, NO_2_ is often accompanied by interfering gases such as H_2_S, CO, and SO_2_. Therefore, sensors with high selectivity and response must be developed to meet NO_2_ detection requirements.

In recent years, resistive gas sensors based on metal oxide semiconductors (MOS) have been widely used for NO_2_ detection because of their low cost, environmental friendliness, and excellent performance. Hydrothermal methods for oxide synthesis are being widely utilized today. Jamnani et al. synthesized Sm_2_O_3_ nanorods through a straightforward hydrothermal process, employing cetyltrimethylammonium bromide (CTAB) as an organic surfactant for detecting volatile organic compounds; they achieved a notable response of 3.41–1 ppm for acetone [5]. Furthermore, flower-like MoS_2_/WS_2_ composites were prepared via a simple one-pot hydrothermal method, enhancing the detection of NO_2_ at room temperature [6]. MOS materials such as WO_3_ [7], SnO_2_ [8], In_2_O_3_ [9], ZnO [10], and CuO [11] are promising for meeting the low-limit, high-response requirements of NO_2_ detection. Additionally, Ag_2_Te, an n-type semiconductor, exhibited commendable NO_2_ sensing performance. The resistance of the Ag_2_Te sensor remained stable in air and at 1 ppm NO_2_ over a humidity range of 10–90%, demonstrating its excellent humidity-resistant properties [12].

WO_3_ is a typical n-type MOS with an adjustable bandgap of 2.5–2.8 eV [13]. Owing to its crystal structure and surface properties, WO_3_ can adopt various microscopic morphologies including nanowires [14], nanospheres [15], nanosheets [16], and urchin-like nanostructures [17]. The microscopic morphology of WO_3_ can be adjusted to strengthen the interactions with gases and increase the possibility of gas adsorption. However, similar to sensors based on other metal oxides, WO_3_ nanomaterial gas sensors exhibit poor selectivity and cannot accurately detect NO_2_ in complex environments in which multiple gases are mixed [18].

MOSs have been doped with different catalysts to improve sensor selectivity and increase the reaction rate between the target gas and sensing layer. Based on electronic sensitization and chemical sensitization mechanisms [19,20], doping with an appropriate amount of a noble metal can improve the gas response of MOS materials. For example, compared with pure ZnO, Pd-modified ZnO nanowires improved response and significantly higher NO_2_ selectivity [21]. As an alternative approach, gas sensors have been prepared using composites of MOSs and reduced graphene oxide (rGO), which has a large surface area and exhibits high electron mobility and high conductivity at low temperatures [22]. Notably, SnO_2_ nanofibers sensitized with rGO nanosheets achieved high response and selectivity for acetone and H_2_S [23]. In addition, recently, phosphorene has garnered research interest owing to its honeycomb structure, high carrier mobility, and tunable bandgap, which ranges from 0.3 eV in bulk to approximately 1.9 eV in monolayers [24]. Black phosphorus (BP), a novel single-element two-dimensional layered material, has been extensively utilized in gas sensing [25]. Han fabricated a BP gas sensor featuring high-quality, few-layered BP micro-ribbons. Energy-dispersive spectroscopy mapping revealed that BP underwent only slight oxidation, achieving low-limit detection and high NO_2_ selectivity under both N_2_ and air conditions [26]. Zhou fabricated a two-dimensional SnO_2_ nanosheet micro-electromechanical system (MEMS) gas sensor decorated with black phosphorus (BP) to detect H_2_S [27]. The BP-SnO_2_ sensor had a lower optimal operating temperature and faster response–recovery times and exhibited higher selectivity than the pure SnO_2_ sensor. When BP, which is a p-type semiconductor, is added to WO_3_, which is an n-type semiconductor, more p-n heterojunctions can be formed. Moreover, the excellent conductivity of BP can increase the carrier mobility of WO_3_. Therefore, BP-WO_3_ composites are of interest for gas-sensing applications.

We fabricated a highly selective BP-WO_3_ gas sensor for NO_2_ detection in this study. BP crystals were prepared via the chemical vapor-phase transport method, and BP-WO_3_ sandwich-like nanostructures were formed by a simple one-step hydrothermal method. The successful preparation of the BP-WO_3_ composite was confirmed through its characterization via scanning electron microscopy (SEM), X-ray diffraction (XRD), and X-ray photoelectron spectroscopy (XPS). Subsequently, a gas-sensing test platform was constructed and the selectivity, response, operating temperature, and concentration range of the BP-WO_3_ sensor were demonstrated to be superior to those of pure WO_3_. This improved sensing performance could be attributed to the enhanced electron-trapping ability of BP-WO_3_.

## 2. Materials and Methods

### 2.1. Preparation of Sensing Materials

BP crystals were prepared via the chemical vapor-phase transport method. First, 0.5 g of red phosphorus, 0.07 g of Sn powder, and 0.03 g of I_2_ were placed in a quartz glass tube with an outer diameter of 18 mm. The quartz tube was then pumped to a vacuum and sealed by fusing with a cylindrical quartz block. The end of the quartz tube containing the raw materials (hot end) was placed in the center of the high-temperature zone of a tube furnace. The heating stage was heated to 600 °C at a rate of 20 °C/h and held for 2 h, cooled to 500 °C at a rate of 10 °C/h and held for 2 h, and finally cooled to 25 °C at a rate of 15 °C/h. The BP crystals formed at the cold end of the quartz tube. The BP crystals were then ground into a fine powder and a 10 mg/mL aqueous solution was prepared.

WO_3_ was prepared via a hydrothermal method. First, 2.05 g of Na_2_WO_4_∙2H_2_O, 0.25 g of Na_2_SO_4_, and 0.3 g of K_2_SO_4_ were added to 40 mL of deionized water. After stirring for 20 min, the pH was adjusted to 1.6 with 2 M HCl. The precursor solution was transferred to a 100 mL Teflon-lined autoclave and heated at 180 °C for 24 h. After cooling to 25 °C, the precipitate was centrifuged at 4000 rpm for 15 min and washed with water and anhydrous ethanol (6 times). Finally, the WO_3_ sample was obtained by vacuum drying at 60 °C for 8 h. The WO_3_ synthesis process is outlined in Figure 1. The BP-WO_3_ composite was synthesized using the same method, except that 1 mL of the aqueous BP solution was added after adjusting the pH.

### 2.2. Characterization of Sensing Materials

The crystal structures of WO_3_ and BP-WO_3_ were analyzed using an X-ray diffractometer (SmartLab 3 kW, Rigaku Corporation, Tokyo, Japan) using Cu Kα radiation in the scanning range of 20°–70°. The valence states of the elements were analyzed using an X-ray photoelectron spectrometer (ESCALAB 250Xi, Thermo Fisher Scientific, Waltham, MA, USA). The morphologies of the samples were observed through field-emission SEM (ZEISS Gemini SEM 300, Carl Zeiss AG, Oberkochen, Germany) at acceleration voltages of 3–15 kV.

### 2.3. Test Platform for Gas Sensing

Sensors were fabricated by coating the sensing material on a MEMS substrate, consisting of an interdigitated electrode, an insulating layer, a heating electrode, a supporting layer, and a silicon substrate (Figure 2a,b). Specifically, the sensing material was ground in an agate mortar to 200 mesh and sieved. A slurry was prepared by uniformly mixing this sample with terpineol. The slurry was coated on the sensing layer of the MEMS substrate, which was then vacuum dried at 65 °C for 5 h and aged at the operating temperature for 1 week. Figure 2c,d show the physical images of the sensor. The test platform comprised a 7.5 L closed air chamber, fan, DC power supply, data-acquisition unit, and PC (Figure 2e). A syringe was used to inject a specific volume of gas into the gas chamber. The fan diffused the gas evenly and quickly into the gas chamber, and the DC power supply provided the working voltage of the sensor. Gas adsorption onto the gas-sensitive material caused a change in the sensor resistance, which was recorded by the data-acquisition unit and then uploaded to the PC.

## 3. Results and Discussion

### 3.1. Material Characterization

Figure 3 shows the XRD patterns of the pure WO_3_ and BP-WO_3_ samples at 25 °C. Both materials are relatively well crystallized and match well with the standard JCPDS card for hexagonal WO_3_ (PDF#97-008-0634), with lattice parameters of *a* = *b* = 7.3244 Å and *c* = 7.6628 Å and diffraction peaks at 13.95°, 23.20°, 24.28°, 28.11°, 33.83°, 36.76°, and 49.75° corresponding to the (100), (002), (110), (200), (112), (202), and (220) crystal planes, respectively. However, because of the extremely low BP content of BP-WO_3_, the XRD pattern of this sample does not contain any diffraction peaks corresponding to phosphorus. Therefore, the samples were further characterized using other methods.

Figure 4a–c show the SEM images of pure WO_3_. The microscopic morphology of pure WO_3_ consists of rod-like structures with diameters of 50 nm and lengths ranging from 100 nm to 1 μm, resulting in a rectangular cross-section. Most nanorods are stacked in parallel rather than dispersed, indicating the presence of strong interaction forces between them. Remarkably, the rod-like structures consist of many fine nanorods (5–10 nm in diameter) assembled into bundles. Energy-dispersive X-ray spectroscopy (EDS) results extracted from the region in Figure 4b show that W and O were uniformly distributed on the nanorods (Figure 4d,e). Based on the characteristic W and O peaks (Figure 4f), the mass fractions of W and O were calculated to be 80.95% and 19.05%, respectively.

Figure 4g–i show the SEM images of BP-WO_3_. The BP-WO_3_ composite was found to possess a novel sandwich-like nanostructure assembled from WO_3_ nanorods and multilayer BP nanosheets. As shown in Figure 4g, the BP nanosheets provide attachment sites for nanowire growth, and both sides of each BP nanosheet are uniformly and densely loaded with nanorods. Notably, the surrounding free WO_3_ nanorods and BP nanosheets do not bind together. According to a previous study, monolayer BP nanosheets have a thickness of 3–5 nm after ultrasonic liquid-phase stripping. In the BP-WO_3_ composite, the nanosheet in the middle position has a thickness of 27.66 nm (Figure 4h), which suggests that this structure consists of 5–9 layers of monolayer BP nanosheets. The magnified SEM images in Figure 4h,j reveal nearly perpendicular nanorod growth on both sides of the BP nanosheets. Moreover, these nanorods are of the same length and cover the nanosheets almost completely. This structure explains why no P was detected through XRD or EDS.

XPS was performed to further investigate the chemical binding states of the samples. Figure 5a shows the XPS profiles of pure WO_3_ and BP-WO_3_. In the WO_3_ sample, the atomic ratio of W to O is 24.02%:75.98%, which corresponds well with the expected atomic ratio (1:3). However, because the XPS beam can only penetrate the sample to a depth of 3–10 nm, the analysis is limited to the sample surface and P was not detected. Figure 5b and c show the W 4f and O 1s XPS spectra of the BP-WO_3_ sample. The W 4f spectrum exhibits two major peaks centered at 37.9 and 35.7 eV [28], corresponding to W^6+^ 4f_5/2_ and 4f_7/2_, respectively. The O 1s spectrum exhibits a major peak at 530.5 eV [29], corresponding to the lattice oxygen (O^2−^) of WO_3_, and a small peak at 532.1 eV [30], ascribed to the chemisorbed oxygen or weakly bonded oxygen species on the WO_3_ nanorods.

### 3.2. Gas-Sensing Properties

The operating temperature is a crucial sensor metric for actual measurement. To explore the effect of the operating temperature on the sensor performance, we performed static measurements using the BP-WO_3_ sensor at 10 ppm NO_2_. Figure 6a shows the variation in the resistance of the BP-WO_3_ sensor over time at temperatures between 190 and 270 °C. The initial resistance of the sensor reaches tens of megahertz and decreases with increasing temperature. The resistance increases significantly after the target gas is introduced and slowly returns to the initial level after releasing the gas. The lowest operating temperature point was selected at 190 °C to avoid large measurement errors caused by high sensor resistance. Figure 6b illustrates the response (R_a_/R_g_, where R_a_ is the resistance of the sensor in air and R_g_ is the resistance of the sensor in the target gas) at different temperatures. Increasing the temperature induces a significant decrease in response; however, the response–recovery times of the sensor become faster. The response of the sensor is only 1.14 at 270 °C but increases to 1.73 at 190 °C. The response–recovery times (defined as the time required for a change of the resistance of 90%) of the sensor are 72 and 92 s, respectively, at 190 °C, but 36 and 36 s, respectively, at 270 °C. The improved response–recovery times of the sensor at 270 °C are favorable for commercial applications. However, as visualization of the sensor response is critical during gas sensitivity tests, we performed further tests at 190 °C.

Figure 6c shows the response of the BP-WO_3_ sensor to NO_2_ concentrations in the range of 2–14 ppm. The sensor response decreases significantly as the gas concentration decreases, indicating an excellent resolving power. In addition, the sensor exhibits fast response–recovery times for NO_2_. Figure 6d shows the relationship between the response of the BP-WO_3_ sensor and NO_2_ concentration in the 2–14 ppm range. The results show an excellent linear relationship (R^2^ = 0.98) with an equation of y = 1.1522 + 0.0613x, implying that the BP-WO_3_ sensor has good calibration capability. Based on the linear fit results, the theoretical BP-WO_3_ response to 10 ppb NO_2_ can be calculated as 1.15.

To verify the NO_2_ selectivity of the sensor, we performed gas sensitivity tests with the BP-WO_3_ and pure WO_3_ sensors using H_2_S as a representative interfering gas. Figure 7a shows the response (R_a_/R_g_) of the pure WO_3_ sensor to 4 ppm H_2_S at different operating temperatures. The sensor response decreases drastically with increasing temperature, but the response–recovery times become significantly shorter. Although the response is higher at 160 °C than at 240 °C, the response–recovery times decrease significantly (from 60 and 100 s, respectively, at 160 °C, to 13 and 15 s, respectively, at 240 °C), consistent with the trend observed for BP-WO_3_. The investigation of the response of the pure WO_3_ sensor to H_2_S at the concentration of 0.3–5 ppm (Figure 7b) reveals good resolving power in the range of 1–5 ppm. However, the difference in response is negligible at concentrations of 0.3 and 0.5 ppm (1.024 and 1.064, respectively), suggesting that the detection limit has been reached. Figure 7c shows the relationship between the response of the pure WO_3_ sensor and H_2_S concentration in the range of 0.3–5 ppm. The results show a good linear relationship (R^2^ = 0.97) with an equation of y = 0.9617 + 0.2225x, indicating the good calibratable capability of the pure WO_3_ sensor. Based on the linear fit results, the theoretical pure WO_3_ response to 10 ppb H_2_S can be calculated as 0.96. In contrast, the BP-WO_3_ sensor exhibited no response to various H_2_S concentrations (Figure 7d), demonstrating that the BP-WO_3_ sensor can effectively avoid the influence of the interfering gas, H_2_S, unlike pure WO_3_, resulting in improved selectivity.

Figure 8a,b show the response curves of BP-WO_3_ and pure WO_3_ over seven cycles at 12 ppm NO_2_ and 3.5 ppm H_2_S, respectively. The response values and response–recovery times of the sensor during each cycle are similar, indicating that the BP-WO_3_ and pure WO_3_ sensors have excellent repeatability. In addition, we tested the responses of the BP-WO_3_ and pure WO_3_ sensors to 1 ppm H_2_S, NO_2_, C_2_H_6_O, CO_2_, and CO (Figure 8b). The pure WO_3_ sensor responded well to both H_2_S and NO_2_, whereas the BP-WO_3_ sensor exhibited almost no response to H_2_S, confirming that the BP-WO_3_ sensor exhibits better NO_2_ selectivity than the pure WO_3_ sensor. The response values of the BP-WO_3_ sensor to NO_2_ and H_2_S over 15 days are shown in Figure 8c. Over this period, the response of the sensor to NO_2_ only decreases slightly, with the retention of more than 95% of the initial performance. Moreover, the sensor remains unresponsive to H_2_S for 15 days, demonstrating the long-term stability of BP-WO_3_. In addition, the gas-sensing properties of previously reported WO_3_-based sensors and the as-prepared system are summarized in Table 1. It can be seen that the BP-WO_3_ sensor demonstrates similar performance to or outperforms other sensors.

### 3.3. Gas-Sensing Mechanism

The gas-sensing mechanism of WO_3_ is similar to that of other MOSs, where charge transfer to the gas during adsorption leads to a change in resistance. When the sensor is exposed to air, oxygen molecules adsorb on the surface of the WO_3_ nanorods and capture free electrons from the WO_3_ conduction band. Thus, the oxygen molecules are converted to the chemisorbed oxygen species ($O_{2}^{-}$ and $O^{-}$) [37], which form an electron-withdrawing layer near the surface of the WO_3_ nanorods. Consequently, the carrier concentration of WO_3_, which is an n-type semiconductor, decreases, leading to an increase in resistance. Upon introducing NO_2_, NO_2_ molecules adsorb on the WO_3_ surface and react with the adsorbed oxygen ions [38]. This process can be described by Equations (1)–(5).
(1)$$O_{2}\left(\mathrm{gas}\right)\to O_{2}\left(\mathrm{ads}\right),$$
(2)$$O_{2}\left(\mathrm{ads}\right)+e^{-}\to O_{2}^{-}\left(\mathrm{ads}\right),$$
(3)$$O_{2}^{-}\left(\mathrm{ads}\right)+e^{-}\to 2O^{-}\left(\mathrm{ads}\right),$$
(4)$${NO}_{2}\left(\mathrm{ads}\right)+e^{-}\to {NO}_{2}^{-}\left(\mathrm{ads}\right),$$
(5)$${NO}_{2}\left(\mathrm{ads}\right)+O^{-}\left(\mathrm{ads}\right)+2e^{-}\to {NO}_{2}^{-}\left(\mathrm{ads}\right)+O^{2-}\left(\mathrm{ads}\right)$$

Doping with BP, a p-type semiconductor, causes most carriers to diffuse on the surfaces of BP and WO_3_, forming numerous p-n heterojunctions. Owing to the difference in the work functions of WO_3_ and BP in the air (5.15 and 3.9 eV [39,40], respectively), electrons are transferred from WO_3_ to the BP surface, while holes move in the opposite direction until a unified Fermi level is formed. An atomic structure model of the BP-WO_3_ heterojunction is shown in Figure 9. When the sensor is exposed to air, the adsorption of oxygen leads to a decrease in the electron density on the surface of WO_3_ and an increase in the hole density of BP, resulting in the formation of an electron depletion layer on the WO_3_ side and a hole accumulation layer on the BP side. As more electrons can be trapped in the material in the presence of NO_2_, the potential barrier of the p-n junctions increases, resulting in a decrease in the conductivity of the material. Although the sandwich-like nanostructure of BP-WO_3_ may contribute to its enhanced selectivity, BP-WO_3_ has a smaller specific surface area than the pure WO_3_ nanorods and thus does not provide enough adsorption sites for the interfering gases. Therefore, the stronger ability of BP-WO_3_ to trap electrons from NO_2_ is likely the main reason for the high selectivity of this material.

Regarding the exceptional selectivity of BP-WO_3_, particularly toward interfering gas H_2_S, several reasons can be posited. First, NO_2_, a nitrogen-based gas, exhibits higher adsorption energy and adhesion coefficient than H_2_S. Previous studies indicate that the adhesion coefficient of H_2_S to phosphorene at 300 K is 0.81, whereas that of NO_2_ is 1.0. Second, the adsorption interaction between NO_2_ and phosphorene induces a more significant alteration in the electronic structure than that achieved with other gases [41]. Finally, the unique BP-WO_3_ sandwich-like nanostructure contributes to its selectivity; the BP nanosheets hinder electron migration in the WO_3_ nanorods, unlike the interaction with whole nanorods, which is less favorable for H_2_S and WO_3_ interaction. However, NO_2_ can extract electrons from both WO_3_ and BP, enhancing its detection.

## 4. Conclusions

BP-WO_3_ sandwich-like nanostructures were fabricated for the first time using a simple one-step hydrothermal method. The applicability of the corresponding BP-WO_3_-based MEMS sensor was applied to highly selective NO_2_ sensing. Unlike pure WO_3_, BP-WO_3_ was largely insensitive to interfering gases, especially H_2_S. In addition, BP-WO_3_ achieved a response of 1.73 for 10 ppm NO_2_ at an operating temperature of 190 °C, response–recovery times of only 40 and 44 s, respectively, at 270 °C, and excellent resolving power and repeatability. The high selectivity of BP-WO_3_ was attributed to the surface oxygen ions having a much higher reaction rate with NO_2_ with than with the interfering gases. The use of the as-prepared BP-WO_3_ composite will allow for the enhancement of the selectivity of gas sensors.

## Figures and Tables

**Figure 1 sensors-24-01376-f001:**
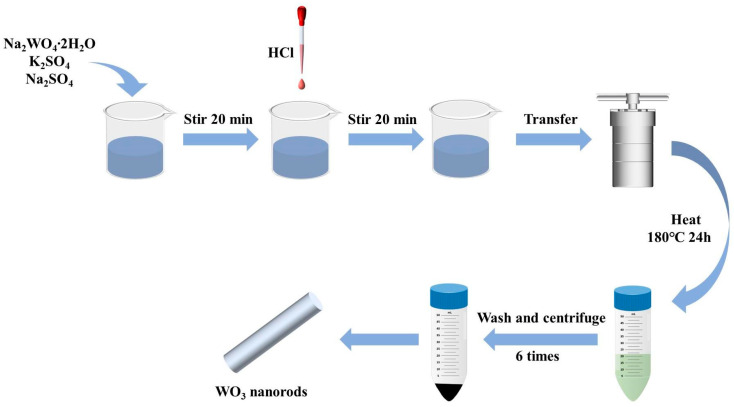
Schematic of the preparation of WO_3_ nanorods.

**Figure 2 sensors-24-01376-f002:**
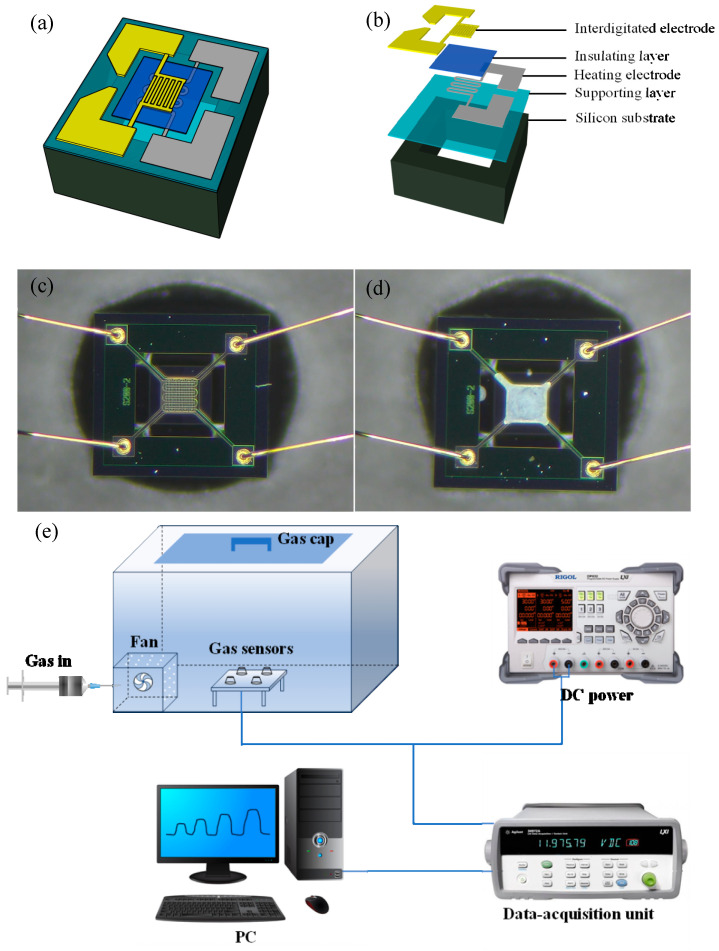
(**a**,**b**) Schematics of micro-electromechanical system (MEMS) structure; images of the MEMS substrate (**c**) before and (**d**) after coating with the sensing material; (**e**) test platform for gas sensing.

**Figure 3 sensors-24-01376-f003:**
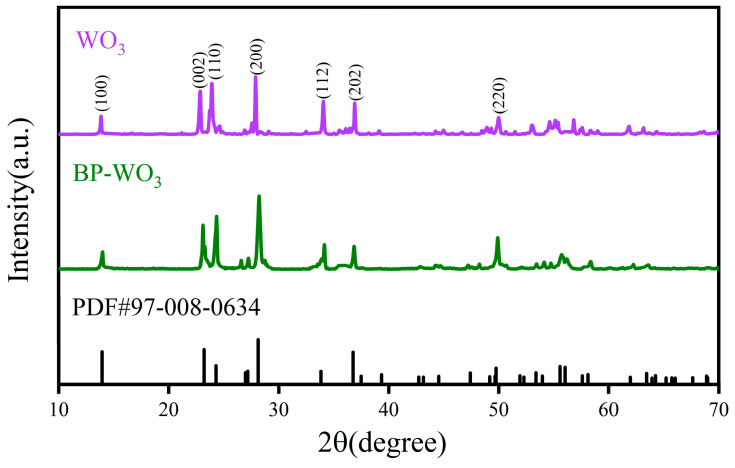
X-ray diffraction (XRD) patterns of pure WO_3_ and BP-WO_3_ at 25 °C.

**Figure 4 sensors-24-01376-f004:**
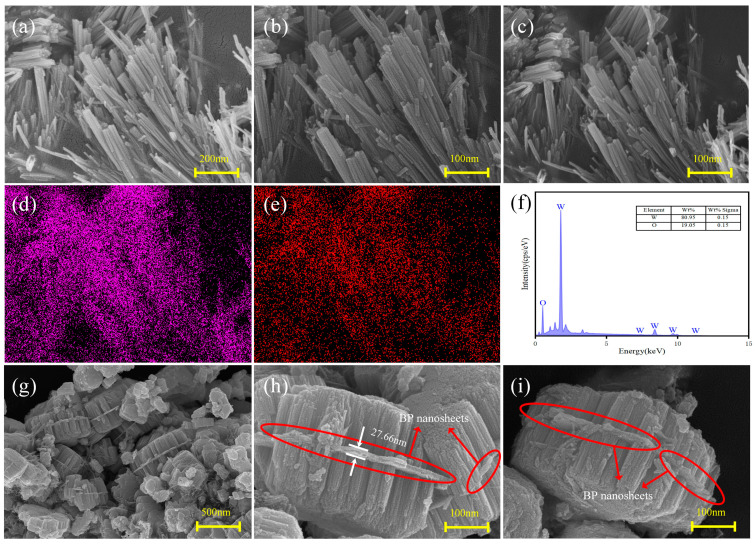
(**a**–**c**) SEM images of pure WO_3_; EDS elemental mapping profiles of (**d**) W and (**e**) O in pure WO_3_; (**f**) EDS spectrum of pure WO_3_; (**g**–**i**) SEM images of BP-WO_3_.

**Figure 5 sensors-24-01376-f005:**
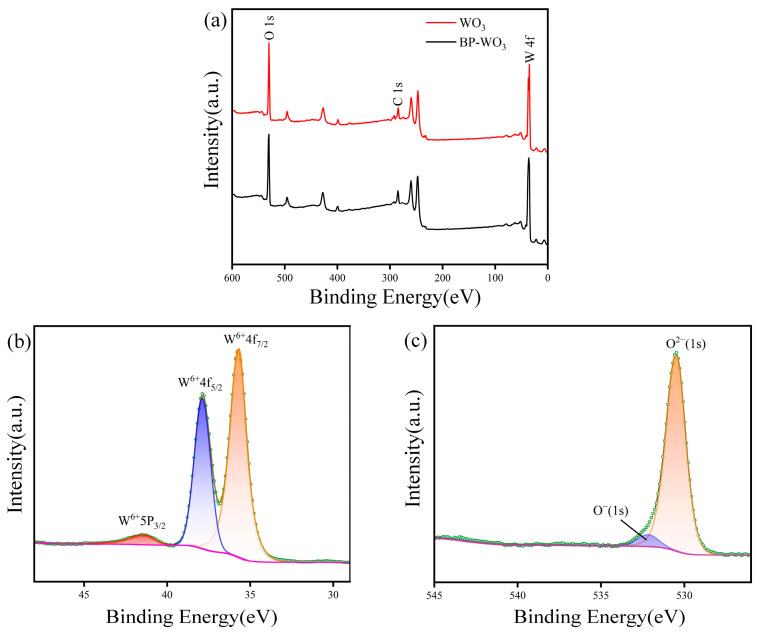
(**a**) XPS survey spectra of pure WO_3_ and BP-WO_3_; (**b**) W 4f and (**c**) O 1s XPS spectra of BP-WO_3_.

**Figure 6 sensors-24-01376-f006:**
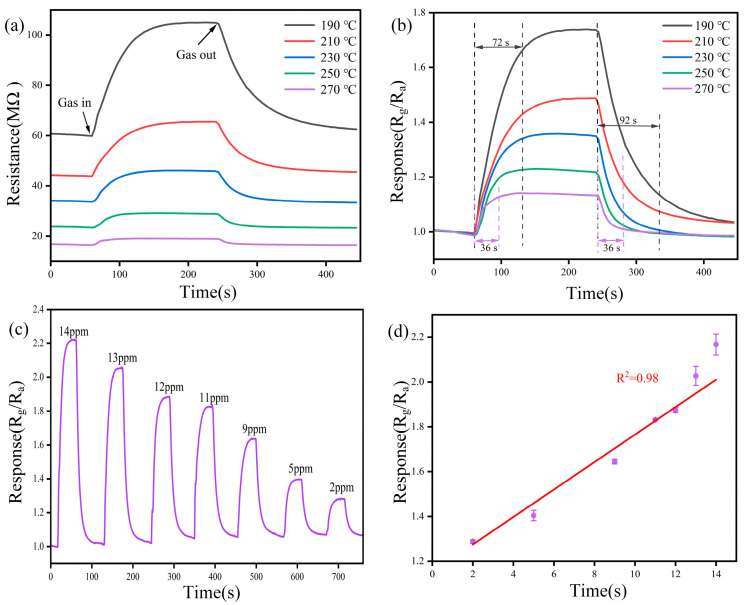
(**a**) Resistance and (**b**) response of the BP-WO_3_ sensor to 10 ppm NO_2_ at various operating temperatures; (**c**) response of the BP-WO_3_ sensor to various NO_2_ concentrations at 190 °C; (**d**) linear fitting of the BP-WO_3_ sensor response as a function of NO_2_ concentration.

**Figure 7 sensors-24-01376-f007:**
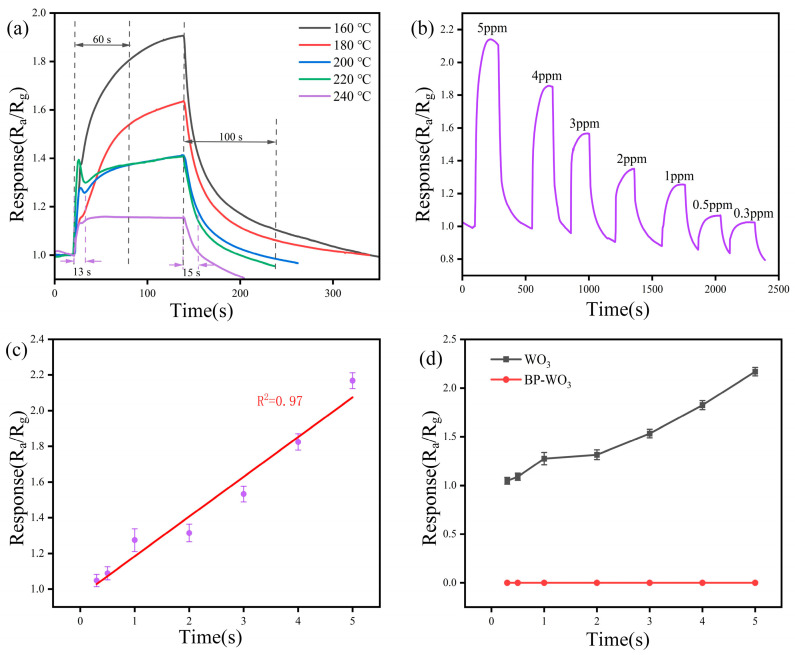
(**a**) Response of the WO_3_ sensor to 4 ppm H_2_S at various operating temperatures; (**b**) response of WO_3_ sensor to various H_2_S concentrations at 160 °C; (**c**) linear fitting of the WO_3_ sensor response as a function of H_2_S concentration; (**d**) response of the WO_3_ and BP-WO_3_ sensors as a function of H_2_S concentration.

**Figure 8 sensors-24-01376-f008:**
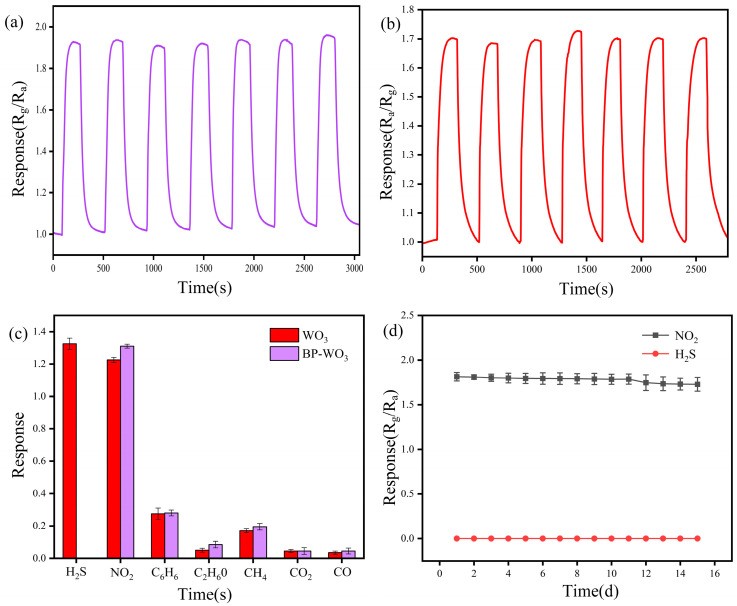
(**a**) Repeatability of the BP-WO_3_ sensor response to 12 ppm NO_2_; (**b**) repeatability of the WO_3_ sensor response to 3.5 ppm H_2_S; (**c**) selectivities of the BP-WO_3_ and WO_3_ sensors for various gases; (**d**) response of the BP-WO_3_ sensor to 12 ppm NO_2_ and 4 ppm H_2_S over 15 days.

**Figure 9 sensors-24-01376-f009:**
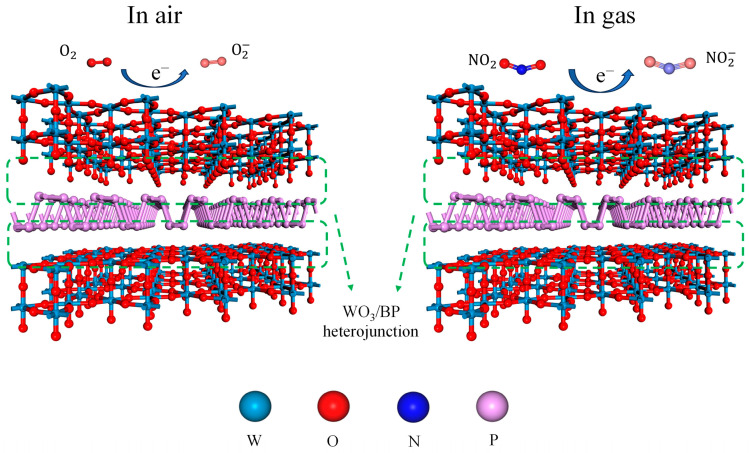
Schematic diagrams of the gas-sensing mechanism of BP-WO_3_.

**Table 1 sensors-24-01376-t001:** Gas-sensing properties of various WO_3_-based NO_2_ gas sensors.

Material	Concentration	Response	Response–Recovery Time	Reference
MWCNT-WO_3_	100 ppm	0.96	10 s/20 min	[31]
MWCNT and rGO-WO_3_	5 ppm	17%	7/15 min	[32]
WS_2_-WO_3_	10 ppm	65	~800/~400 s	[33]
Fe-WO_3_	12 ppm	105%	250/650 s	[34]
PPy-WO_3_	5 ppm	12%	~370/~50 s	[35]
WO_3_	10 ppm	2.02	96/81 s	[36]
BP-WO_3_	14 ppm	2.21 (122%)	72/92 s	This work

## Data Availability

Data are contained within the article.

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
