# Peer review of "Black Phosphorus–Tungsten Oxide Sandwich-like Nanostructures for Highly Selective NO2 Detection"

_sensors, 2024, doi:10.3390/s24051376_

Round 1

Reviewer 1 Report

Comments and Suggestions for Authors

see attached file.

Comments on the Quality of English Language

Minor editing of English language required

Reviewer 2 Report

Comments and Suggestions for Authors

In this paper (sensors-2878442), the authors exhibited a BP-WO3 gas sensor for NO2 detection. The methods and results are acceptable, however, there are some problems in the motivation, experiment, results and discussions. Therefore, some revisions are needed. My specific comments are listed below:

1. Abstract and results: The response and sensitivity of gas sensors should not be confused. In fact, the data (1.25–2.21) mentioned by the authors is response values. Sensitivity is defined as a linear fitting coefficient of response within a certain gas range.

2. Introduction: “BP-SnO2”: The numbers in the chemical formula require subscripts, including references.

3. Introduction: Why choose BP s as a modification material? In fact, BP itself also has a good NO2 gas sensing response. It is recommended to supplement the discussion on BP NO2 gas sensors. How to overcome the issues of thermal stability and oxidation of BP?

4. Materials and Methods: How is the material ratio of SnO2 and BP determined?

5. Materials and Methods: How to configure different concentrations of NO2?

6.  “…the sensitivity (Ra/Rg….”). Response rather than sensitivity.

7. The parameter definitions of gas sensors need to be provided, such as sensitivity, and response/recovery times.

8. To evaluate the gas sensing performances, it is recommended to compare it with relevant reports.

9. Check the format of the references one bye one, such as abbreviating the journal name. In addition, the development of NO2 sensors is rapid, and most of the references are outdated. It is recommended to cite some recent literature on NO2 sensors, such as Sens. Actuators, B Chem. 2022, 363, 131790.

Comments on the Quality of English Language

Minor editing of English language required

Round 2

Reviewer 1 Report

Comments and Suggestions for Authors

The authors answered all questions. I recommend their publication.

Reviewer 2 Report

Comments and Suggestions for Authors

The response and revised manuscript are satisfactory, and it is recommended to accept.